# Effect of Graphene Oxide Surface Deposition Process on Synthetic Macrofibers and Its Results on the Microstructure of Fiber-Reinforced Concrete

**DOI:** 10.3390/polym16081168

**Published:** 2024-04-21

**Authors:** Vinício Cecconello, Matheus Poletto

**Affiliations:** Postgraduate Program in Engineering of Processes and Technologies (PGEPROTEC), University of Caxias do Sul (UCS), Caxias do Sul 95070-560, Brazil; vcecconello@ucs.br

**Keywords:** synthetic macrofibers, graphene oxide, fiber-reinforced concrete

## Abstract

The improvement of the mechanical properties of concrete can be achieved with the use of synthetic macrofibers. However, this fiber–matrix interaction will be sufficiently efficient for tensile efforts only when there is a binding agent that associates the characteristics of the paste with the characteristics of the surface of the reinforcing material. As already identified, in a first phase of this research using synthetic microfibers, a better fiber–matrix interaction can be achieved with the surface treatment of synthetic fibers with graphene oxide. In this way, we sought to evaluate the surface treatment with graphene oxide on two synthetic polypropylene macrofibers (macrofiber “A” and macrofiber “B”) and its contribution to the concrete transition zone. The surface deposition on the macrofiber was carried out using the ultrasonication method; then, the macrofiber with the best deposition for creating reinforced concrete mixtures was identified. To evaluate the quality of GO deposition, scanning electron microscopy (SEM-FEG) and energy-dispersive spectroscopy (EDS) tests were carried out; the same technique was used to evaluate the macrofiber–matrix transition zone. The SEM-FEG images indicated that macrofiber “B” obtained greater homogeneity in surface deposition and it presented a 13% greater deposition of C in the EDS spectra. The SEM-FEG micrographs for reinforced concrete indicated a reduction in voids in the macrofiber–matrix transition zone for concretes that used macrofibers treated with GO.

## 1. Introduction

Concrete and cement-based products used in civil construction commonly exhibit brittle behavior, which reduces their durability, causing significant losses to the industry and its users. Considering that these products are among the most consumed in the world, the importance of proposing improvements to increase their useful life when used is notable.

Some authors have focused their research on discussions regarding mechanical properties, the effects of the fiber volume fraction, and variations in the type of fibers used, such as carbon fibers, polypropylene fibers, aramid fibers, and basalt fibers [1,2].

However, research evaluating surface treatment on macrofibers, as proposed by this study, remains rare. Some improvements have already been identified in previous research conducted by the same authors when evaluating concrete microstructures using microfibers treated with graphene oxide. This study advances by discussing the effect of graphene oxide when used with synthetic macrofibers.

The addition of polymeric fibers in cementitious composites is associated with the improvement of their properties against tensile stresses, whether these stresses are of mechanical origin or due to the intrinsic characteristic of the material caused by water loss in the mixtures. Increased tensile strength results in more durable materials with fewer instances of pathological manifestations, as small cracks can become pathways for accelerated degradation.

The use of polymeric fibers positively influences the mechanical properties of the material. However, in the fresh state, they increase yield stress and viscosity, significantly affecting rheological properties. This effect varies depending on the fiber shape, dimensions (aspect ratio), fiber constituent material, volume, and hydrophilic characteristics [3]. Therefore, for the proper production of cementitious composites reinforced with polymeric fibers, understanding the interaction between the matrix and the fibers to be used is crucial, as it is directly associated with the quality and properties of the composite generated.

The use of polymeric fibers involves adding chemically inert materials that remain unreactive in concrete, relying solely on friction between individual fibers and the surrounding concrete matrix to resist applied loads [4]. Shafei et al. [4] also emphasize that the hydrophobic nature of polymeric fibers often results in globally weak fiber–matrix interactions. It is expected that in fiber-reinforced cement-based mixtures, weak areas may form near the fiber–matrix transition zone due to the “wall effect” caused between the materials, a phenomenon that can vary due to fiber composition [5]. Noh et al. [6] also stress the importance of controlling the fiber content in the mixture, as a higher fiber volume leads to weaker fiber–matrix interaction, significantly altering the mixture due to changes in the fiber inclination angle and dispersion in the cementitious matrix [7].

He et al. [8], analyzing results from the fiber–matrix transition zone, identified porous regions with thicknesses between 100 and 150 μm. They also identified heterogeneity in pore distribution between the matrix and the fiber, with higher porosity near the fiber region, about 50%, significantly reducing the contact area with the cement paste.

The fiber–matrix interaction can be purely mechanical; if the concrete matrix strength is high, or adequate mechanical anchoring is provided to the fibers (through geometric modifications), the failure mode may shift to individual fiber rupture using their full capacity. Alternatively, mechanical and chemical interactions can be induced. When a reaction occurs between the fibers and the concrete matrix, chemical bonding assists in friction resistance, providing a combination of bonds. Fiber–matrix interaction characteristics are particularly important in fiber-reinforced composite products because they directly affect the mechanical properties pre- and post-cracking [4].

This enhancement with combined interactions can be achieved by coating the polymeric fibers and altering their hydrophobicity by incorporating elements that generate strong interactions with the cementitious matrix, such as graphene (G) and graphene oxide (GO). Wu, Qureshi, and Wang [9] indicate that the most effective way to use graphene in cement-based products is through the surface coating of fibers. This can enhance the mechanical properties of the cementitious composite, strengthening the interface between the cement matrix and the fiber through GO coating, and enabling the integration of sustainability, multifunctionality, and durability [10].

Lu et al. [11], comparing the properties of cementitious composites reinforced with GO-treated polymeric fibers, found a 46.3% improvement in tensile strength and a 70.4% increase in deformation capacity. They also observed that the average crack width could be reduced from 138 μm to 58 μm.

Among the variables for producing graphene-oxide-treated fiber-reinforced concrete, the deposition method must be considered. The higher the amount of GO sheets dispersed and deposited on the fiber surface, the better the interaction with the cementitious matrix [12].

Utilized from [10,11,13,14], the ultrasonication method is one of the most accepted in this research field. It involves the mixture of graphene oxide and a superplasticizer additive in an aqueous medium, initially mechanically stirred (150 rpm), followed by the addition of synthetic fibers and subjected to ultrasonication for 6 h, ensuring solution stability, with a temperature increase of up to 80 °C. However, no methods are identified in the literature that are suitable for use on synthetic macrofibers.

Therefore, the objective of this work is to evaluate the ultrasonic surface treatment method adopted by Cecconello and Poletto [12] for use on synthetic macrofibers and the effect of these macrofibres on the microstructure of reinforced concrete (FRC). The depositions of GO on the surface of the microfibers and the microstructure of the FRC will be evaluated using SEM-FEG and EDS.

## 2. Materials and Methods

The materials used to develop the research and the methods adopted for the surface deposition of graphene oxide on the surface of synthetic macrofibers, as well as the production of concrete reinforced with macrofibers, will be explained below.

### 2.1. Graphene Oxide

Graphene oxide (GO) was synthesized previously by modifying the Hummers–Hofmann method, starting from Micrograph 99835HP powder graphite (National Graphite Company, São Paulo, Brazil). This process yielded nanosheets consisting of 75% carbon and 21% oxygen.

### 2.2. Coupling Agent

The coupling agent utilized was sourced from Sigma Aldrich and comprises octanoic acid. It has a density of 0.91 g/cm^3^ at 20 °C and a boiling point of 237 °C.

### 2.3. Additive

The superplasticizing additive used is a third-generation polyester carboxylate solution in an aqueous medium, supplied by the Sika Company, São Paulo, Brazil.

### 2.4. Macrofibers

The polymeric fibers selected for use are readily available on the domestic market and are recommended for concrete applications. Table 1 presents the information available from manufacturers. The melt flow rate (MFR) values for both fibers were obtained using a plastometer operating at 230 °C and with a load of 2.16 kg, according to ASTM D1238 [15]. Based on the results presented in Table 1, macrofiber B presented higher viscosity than macrofiber A, possibly due to its higher molar mass.

### 2.5. Fiber Treatment

The surface deposition treatment with graphene oxide (GO) in this study involved an application to synthetic macrofibers, together with a coupling agent and a third-generation superplasticizing additive.

Initially, a mixture consisting of 800 g of water, 3.2 g of GO, and 0.5 g of the superplasticizing additive was prepared and mechanically stirred for 5 min at 500 rpm in a vertical mixer.

The amount of GO was used following the GO mass/fiber mass ratio recommended for carrying out the treatment on microfibers [10,11], considering that this is the first stage of study and new treatment conditions will be listed in future studies.

Subsequently, the solution container, together with the mechanical stirrer, was connected to an ultrasonication device (Bransonic^®^ MH 2800 from Branson, MO, USA, operating at a frequency of 40 kHz), where mechanical agitation and ultrasonification were carried out simultaneously for 30 min.

In a separate container, 9 g of octanoic acid was added to 18 g of fiber and mixed by hand until the octanoic acid coated the entire surface of the fiber. After pretreatment with octanoic acid, the fibers were introduced into the mixture of water, GO, and superplasticizer additives. This combined solution was subjected to mechanical stirring (250 rpm), ultrasonication (40 kHz), and temperature elevation to 80 °C for 6 h.

After the mixing process, the samples were removed, placed on a tray, and dried in an oven with air circulation at 80 °C for 24 h.

### 2.6. Production of Macrofiber-Reinforced Concretes

In the production of reinforced concrete with treated and untreated macrofibers, materials readily available in Brazil were employed, such as Portland Type V cement, natural fine aggregate (with a maximum size of 1.18 mm and a fineness modulus of 1.38), crushed fine aggregate (with a maximum size of 4.75 mm and a fineness modulus of 2.75), a superplasticizer additive, and water.

The objective was to investigate the interaction between the fibers and the concrete matrix. To achieve this, the composition and proportions of the materials followed the recommendations from the literature for the fiber pull-out test, as described in studies [16,17,18].

The water-to-cement ratio and the quantity of superplasticizer additive were determined according to guidelines from the literature, aiming to achieve an appropriate consistency for molding the specimens, as evaluated by the mini-slump test, within a range of 150 mm ± 5 mm. The quantity of materials used for impressions is shown in Table 2.

For the two traits, the macrofibers with and without treatment were deposited during the molding process following the recommendations [17,18]. Initially, the test specimens were prepared, and the fibers were positioned in the mold, with the help of paper with a weight of 300 g/m^2^ and a thickness of 1 mm, to fix them and guarantee their positioning. The fluid concrete was then placed on one side of the form and the embedded position of the fiber was checked so that it was 24 mm on both sides.

After 24 h, molding on the opposite side was carried out and the molding was completed. The specimens were protected with plastic film for another 24 h in a humid chamber; then, they were removed from the molds and placed in submerged curing for 28 days, counting from the last molding to carry out pull-out tests. Subsequently, the specimens were prepared to perform the SEM-FEG and EDS tests to characterize the macrofiber–matrix interface.

### 2.7. Assessment of Surface Deposition on Synthetic Macrofibers

The interaction between graphene oxide and the fiber, as well as the interaction between the fibers (with and without treatment) and the cementitious matrix, was evaluated using SEM-FEG and EDS.

The SEM-FEG analysis was carried out at the Central Microscopy Laboratory (LCMIC) of the University of Caxias do Sul (UCS) using a Tescan scanning electron microscope—model FEG Mira 3 (Brno-Kohoutovice, Czech Republic).

To carry out the SEM-FEG analysis, the fibers were separated, seeking not to interfere with the surface treatment carried out, and sent for analysis, providing the contents of the main components of each sample, and allowing the C and O ratio of each one to be identified.

For the SEM-FEG analysis of cementitious composites, the samples were prepared with dimensions of 1.0 × 1.0 × 0.5 cm (width × length × height), keeping the cracked face in the tensile strength test (which was and will be analyzed) without processing to enable the analysis of the interface of the cement matrix with the fibers and the GO.

The samples were left in a desiccator with moisture-absorbing material for 5 days.

## 3. Results and Discussion

The SEM-FEG and EDS results are presented below to evaluate the surface treatment of the macrofibers used to identify the fiber with the best performance in terms of deposition. Subsequently, the SEM-FEG and EDS results are presented for concrete reinforced with macrofibers that showed the highest GO deposition.

### 3.1. Scanning Electron Microscopy (SEM-FEG) of the Fibers

In Figure 1, showing the surface of macrofiber A, fiber filaments are observed with magnifications of 50× ((a) and (d)), 250× ((b) and (e)), and 1000× ((c) and (d)), comparing samples without surface treatment ((a), (b), and (c)) to those with surface treatment with GO ((d), (e), and (f)).

Figure 1 presents the morphology of sample macrofibers A, with higher magnifications (Figure 1a–c) highlighting the filaments of the fiber in its commercial form, untreated. The linearity of the sample, characteristic of the extrusion process, can be observed in image (Figure 1c), where granular elements can also be identified, which may indicate some deficiency in the fiber production process, apparently due to contamination of the samples. Cardoso and Andrade [19] mention that granular products identified in polymers are an indication of sample contamination during the production process; this can influence the reduction of the treatment area and loss of adhesion properties with the matrix.

In micrographs Figure 1d–f, magnifications of macrofibers A after treatment are presented. It can be observed that the deposition of angular-shaped flake-like structures are identified as GO sheets on the surface of the fiber, in addition to thread-like structures arising from the treatment process, since, during the process, the fiber swells, which can disrupt surface fibrils. Therefore, what can be observed in the micrographs are these fibrils deposited next to the GO.

It can be observed that the deposition of GO on the fiber surface leads to a change in texture. In the micrograph of Figure 2f, it is possible to identify the GO nanosheets and their heterogeneity along the filament surface.

In Figure 2, magnifications of macrofibers B without treatment (Figure 2a–c) and with surface treatment (Figure 2d–f) are presented.

In micrographs Figure 2a–c, the magnifications for macrofibers B without treatment are shown. The continuity of the fiber filament is observed, and longitudinal grooves and fissures are also identified on its fiber surface; it is now possible to identify an already defibrillated part of the macrofiber in Figure 3c.

The micrographs in Figure 3d–f identify macrofibers B after carrying out the treatment; in enlargements (Figure 2d,e), the continuity of the GO coating layer and wrinkles on the fiber surface are observed. In magnification (Figure 2f), the oxide nanosheets and their distribution on the surface of macrofiber B are identified, presenting a favorable condition for interaction with the cementitious matrix, which could contribute to increased density in the region and better fiber–matrix interaction. The thread-like structures identified in Figure 2 can also be observed after treatment, understanding that macrofiber B defibrillation also occurred during treatment, even more so, as the rupture of fibrils has already been identified in Figure 2c.

Enhancing the deposition of graphene oxide (GO) on the fiber surface is crucial for improving its interaction with the cement paste in fiber-reinforced concrete and mortars. This heightened deposition accelerates cement hydration, as indicated in [20], by augmenting the density of cementitious composites in the region containing nanoparticles, as proposed by Sanglakpam and Rizwan [21]. The binding effect and densification of the matrix are directly linked to the quality of Portland cement hydration products, consequently yielding products with superior strength and durability.

### 3.2. Energy-Dispersive Spectroscopy (EDS) of Fibers

In Figure 3, the results for macrofiber A are presented in samples without surface treatment and with surface treatment with GO at different points on the sample surface.

**Figure 3 polymers-16-01168-f003:**
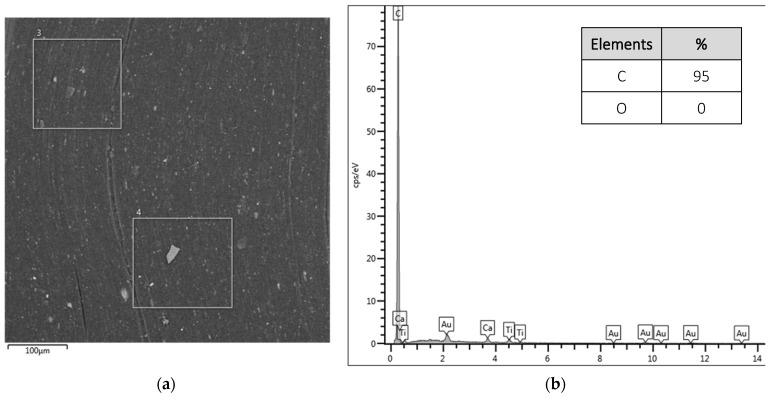
EDS results for macrofibers A: (**a**) micrograph of the fiber surface without treatment highlighting the analyzed points, (**b**) elements that make up the analyzed points for the fiber without treatment, (**c**) micrograph of the fiber surface with treatment highlighting the analyzed points, and (**d**) elements that make up the points analyzed for the treated fiber.

Micrographs Figure 3a,b demonstrate the surface of the untreated sample and the results for the points selected for the spectrum. A sample is observed with homogeneous elements and is predominantly made up of C (95%), since the polypropylene that constitutes the fiber does not have oxygenated groups in its constitution.

In the micrographs of fibers with treatment (Figure 3c,d), differences can be observed between points 1 and 2 of the spectrum. Thus, the heterogeneity of the surface already identified in the FEG-SEM micrographs is confirmed. In spectrum 1, the C/O ratio is 2.8; in spectrum 2, the composition of the elements is identical to the macrofiber without treatment.

This surface heterogeneity, presenting smooth and rough surfaces, may harm future applications in reinforced concrete. As already indicated by Singh, Shukla, and Brown [22], polypropylene fibers have a weak bond with the cement matrix due to their smooth surface, which does not allow for the development of sufficient friction with the matrix; thus, the surface modification of the fibers contributes to a greater bond.

With an effective deposition of the nanoparticle on the surface of the fibers, when introducing it into the concrete mix, the hydration of the cement is accelerated due to the seeding effect of GO clusters that provide nucleation sites due to their small particle size and large surface area [20]. Evaluation of the microstructure of these mixtures suggests that these nanomaterials generate an increase in the diameter and volume of pores in the gel phase, which are normally between 40 and 200 nm [23,24], which may favor the migration of Ca to the surface, changing the Ca/Si ratio. Still, Chougan et al. [24] add that the result may be associated with the interaction between the hydration phases of cement and GO that promote the hydration process, changing the microstructural characteristics of C-S-H.

For Figure 4, the results for macrofibers B are presented in samples without surface treatment and with surface treatment with GO at different points on the sample surface.

Figure 4a,b present the results for macrofiber B without treatment at the points of the spectrum that demonstrate homogeneity in the distribution of the constituent elements, having an average of 99% C, characteristic of a non-recycled polymer.

In the spectra of the treated fibers, shown in Figure 4c,d, the presence of O is observed in the samples, resulting in an average C/O ratio equal to 3.22, demonstrating a better interaction with the surface of the polypropylene fiber as it presents a greater amount of C on its surface.

In Figure 4a, smaller grooves can be seen on the surface of the macrofiber, which possibly contributed to the deposition of GO during the treatment. According to Lamba, Raj, and Singh [25], these small cracks on the surface of the fibers have the potential to strengthen the interface with the matrix, which can contribute to the force versus displacement stages of reinforced concrete, especially in the post-cracking stages when friction is of greater importance [26].

The SEM-FEG and EDS analyses indicated the best results for macrofiber B, showing better homogeneity in the distribution of GO on its surface and better quantitative data with the percentage of GO deposited, and demonstrating promising results for its application in concrete, potentially contributing to the production of more durable materials, and being selected for continued research as an application in structural concrete in this stage of research.

### 3.3. Scanning Electron Microscopy (SEM-FEG) of Concrete Reinforced with Macrofiber

After the quality deposition evaluation trials, which selected the best results for macrofiber B, reinforced concretes were produced using only this macrofiber, as there was no longer a need for distinction in the subsequent results. Following the pull-out test, samples were prepared for SEM-FEG and EDS analysis to identify the main elements and distribution of the analyzed macrofiber-reinforced concretes. The SEM-FEG micrographs of the untreated and treated macrofiber-reinforced concrete are presented in Figure 5.

In micrographs Figure 5a–c, it is possible to identify the concretes produced with the macrofiber without surface treatment with graphene oxide. The fraying of the macrofiber is noticeable, which may be caused by the pull-out tension generated during the test, the formation of voids between the macrofiber strands in (Figure 5c), and the formation of voids between the fiber and matrix (Figure 5c), apparently formed due to the difficulty of interaction between the paste and the fiber.

This can reduce the stages of behavior before and after cracking in the first stage of the pull-out force of concrete reinforced with macrofibre, since, in the first stage, the force increases proportionally to the displacement, demonstrating a linear increase by Hook’s law of elastic deformation. With the appearance of the crack, the second stage begins, where the slope of the force–displacement curve changes; in this phase, the crack is almost proportional to the displacement, and the slope of the curve in this region is a function of the frictional stress in the interface region. In the third stage, the crack propagation is no longer proportional to the displacement and presents instability in the curve, caused by the transfer of stress in the remaining part connected to the fiber surface. In the fourth and final stage, only the friction forces remain within the channel created between the fiber and the matrix. Depending on the friction coefficient of the cement matrix and the fiber polymer, the force may decrease or increase until complete detachment [27,28].

Garcia-Diaz et al. [16] also identified misalignments of the synthetic fibers used after the pull-out test and concluded that this effect is due to the way they are shaped. Most synthetic fibers are made up of filaments, and these behave like individual fibers within the concrete, including with different interactions.

Images Figure 5d–f present the concretes reinforced with the treated macrofiber. Surfaces with greater homogeneity are identified, with a lower number of voids between the fiber and the paste, suggesting that there was a better fiber–matrix interaction due to the presence of GO on the macrofiber surface. Like what was observed in the evaluation of microfibers and as recognized in the literature, a similar effect can be achieved for macrofibers [12,29,30].

Li et al. [31] verified improvements in mixtures due to physical and chemical effects with the addition of calcium carbonate nanoparticles, contributing to the fiber–matrix transition zone, and reducing the propagation of cracks at the macro- and microscopic levels.

The interaction between synthetic fibers and the matrix was also discussed by López-Buendía et al. [32], who identified improvements in the polypropylene matrix–fiber connection with an increase in the surface roughness of the fibers, including with cement hydration elements on these surfaces, as can be seen in Figure 6.

### 3.4. Energy-Dispersive Spectroscopy (EDS) of Concrete Reinforced with Macrofiber

Next, in Figure 6, the EDS results are presented with the analyzed images of the reinforced concretes and five spectra of each sample with the main identified elements.

In Figure 6a,b, spectra and their respective points are identified for the sample of fiber-reinforced concrete with treatment (FRC). For spectra of points 1, 2, 3, and 5, located on the fiber, a more balanced concentration of C, O, and Ca is presented, while for spectrum 4 located near the matrix, the highest percentage of O and the lowest percentage of C are observed.

When analyzing the sample of concrete reinforced with fiber and graphene oxide (MFRC/GO), it was identified that the points on the fiber and at the interface (points 6, 7, 8, and 10) present a high concentration of C and O, elements present in fiber and GO.

At point 9, located next to the matrix, a high concentration of Ca is identified, resulting from the hydration of the cement. Lamba, Raj, and Singh [25] indicate that crystalline silicon and calcium are identified together with the cement-based matrix, with hydrated calcium silicate and calcium hydroxide being the main hydration products.

Comparing the samples (MFRC and MFRC/GO), it is possible to observe the densification of the elements generated during cement hydration, resulting in lower porosity and better pore distribution in the region, which will lead to an improvement in the macrofiber–matrix interaction, as already reported by Wang, Jiang, and Wu [33] when using GO and also previously observed when using microfibers by Cecconello and Poletto [12]. Thus, the GO present on the fiber surface contributes to reducing voids at the macrofiber–matrix interface and aids in cement hydration, as evidenced by the presence of Ca in the region.

The densification of the cementitious composites was also observed by Sanglakpam and Rizwan [21], where varying amounts of GO were added, resulting in considerable improvements in porosity and pore size distribution, culminating in a mechanical and chemical interaction between the matrix and its reinforcement, providing a combination of bonds and directly affecting mechanical properties [4]. Li et al. [31] highlight that with densification and improvement in fiber–matrix interaction, there will be a reduction in the propagation of cracks at the macro- and microscopic levels, as well as an improvement in tensile strength and toughness and, consequently, in the durability of cementitious materials.

## 4. Conclusions

This research sought to evaluate the effects of using graphene oxide as a deposition on synthetic fibers for use in fiber-reinforced concrete, comprising a second stage of the method already adopted by the authors. At the end of this stage, it is possible to better understand the macrofiber deposition method and the interactions among the macrofiber–GO–matrix in the reinforced concrete produced.

Regarding the evaluation of graphene oxide deposition on synthetic macrofibers, it is possible to identify that there are differences in deposition depending on the characteristics of the fiber polymer. The macrofiber with the highest C content in its composition showed greater surface homogeneity and a higher C/O ratio (13% higher).

The microstructure of concrete reinforced with macrofibers indicated changes with GO, such as a reduction in voids at the macrofiber–matrix interface, increasing the densification of the region, and contributing to the fiber–matrix interaction.

## Figures and Tables

**Figure 1 polymers-16-01168-f001:**
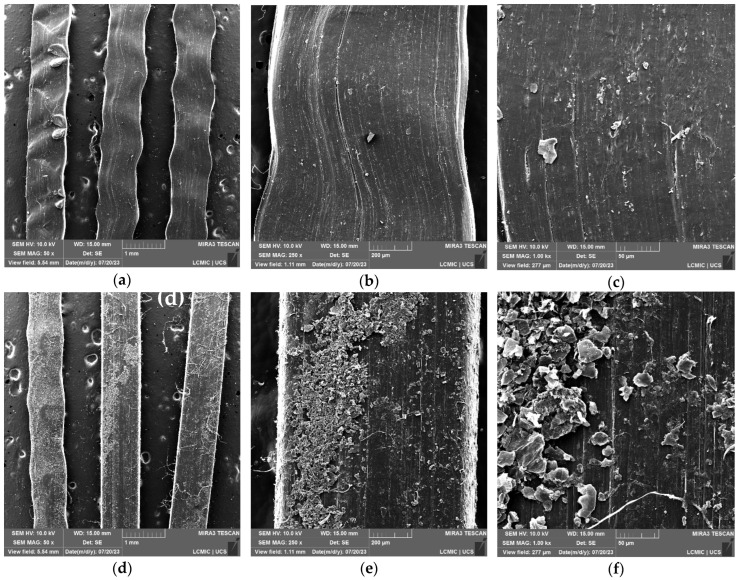
SEM-FEG results of macrofibers A: (**a**) without treatment at 50× magnification; (**b**) without treatment at 250× magnification; (**c**) without treatment at 1000× magnification; (**d**) with treatment at 50× magnification; (**e**) with treatment at 250× magnification; (**f**) with treatment at magnification of 1000×.

**Figure 2 polymers-16-01168-f002:**
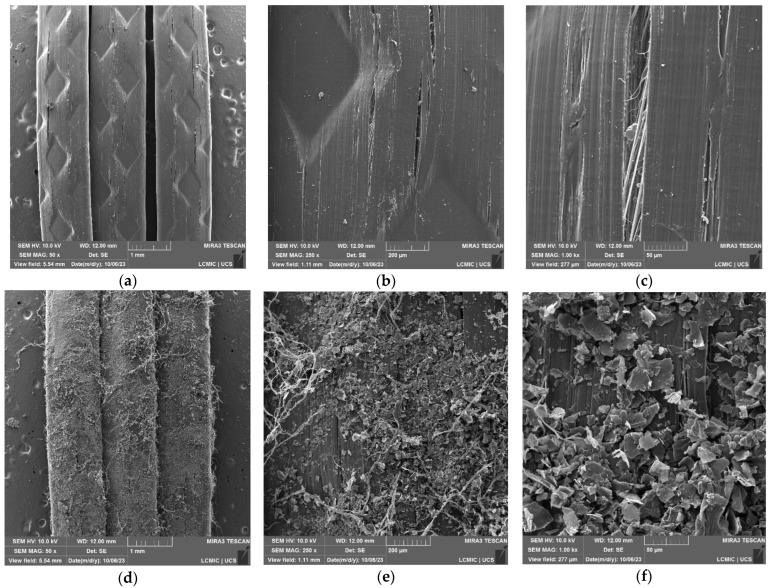
SEM-FEG results for macrofibers B: (**a**) without treatment at 50× magnification; (**b**) without treatment at 250× magnification; (**c**) without treatment at 1000× magnification; (**d**) with treatment at 50× magnification; (**e**) with treatment at 250× magnification; (**f**) with treatment at magnification of 1000×.

**Figure 4 polymers-16-01168-f004:**
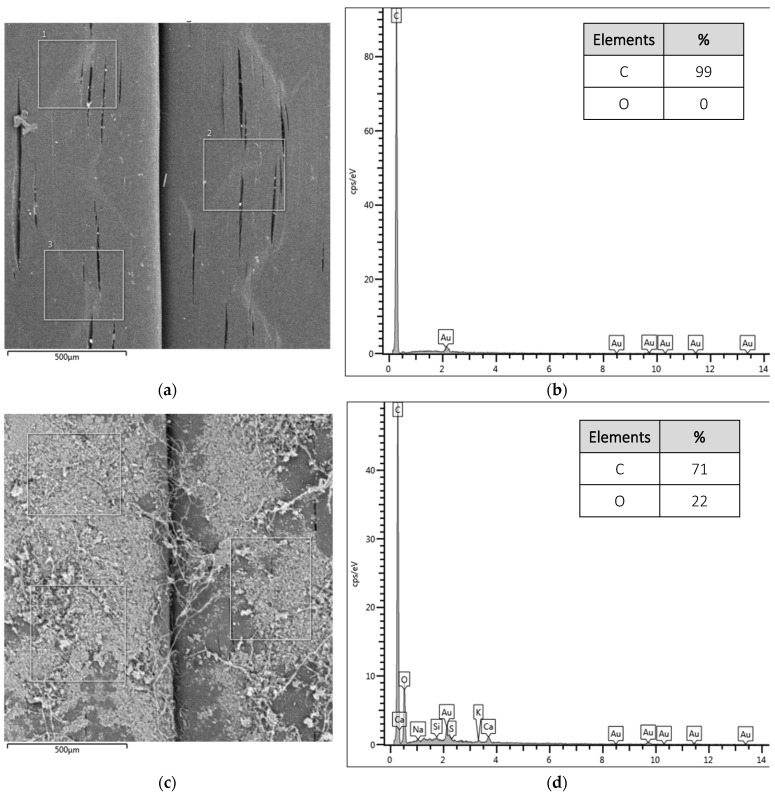
EDS results for F3: (**a**) micrograph of the fiber surface without treatment highlighting the analyzed points, (**b**) elements that make up the analyzed points for the fiber without treatment, (**c**) micrograph of the fiber surface with treatment highlighting the analyzed points, and (**d**) elements that make up the points analyzed for the treated fiber.

**Figure 5 polymers-16-01168-f005:**
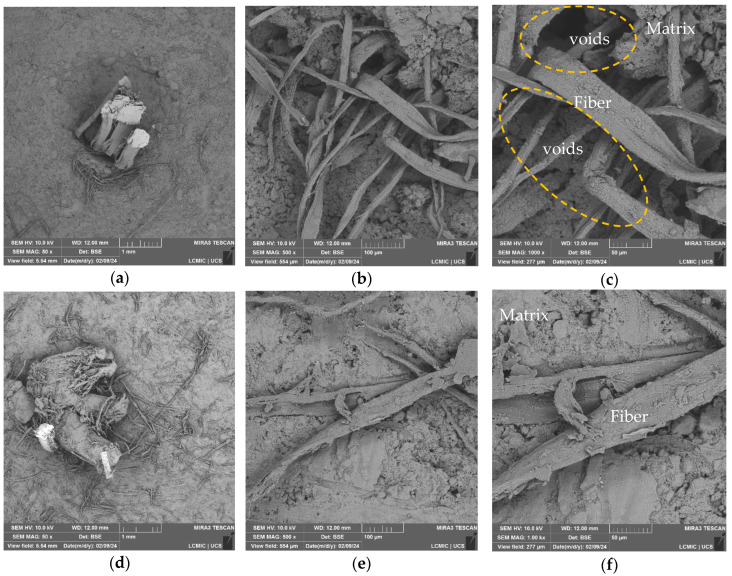
SEM-FEG results of concrete reinforced with macrofibers without treatment (FRC) and with treatment (FRC/GO): (**a**) without treatment at 50× magnification; (**b**) without treatment at 500× magnification; (**c**) without treatment at 1000× magnification; (**d**) with treatment and 50× magnification; (**e**) with treatment and 500× magnification; (**f**) with treatment and magnification of 1000×.

**Figure 6 polymers-16-01168-f006:**
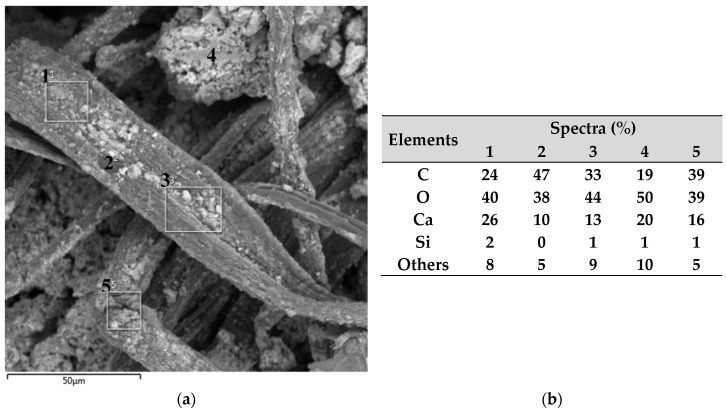
Identification map of spectra and their elements. (**a**) Untreated macrofiber-reinforced concrete (FRC). (**b**) Elements of the spectra for FRC. (**c**) Macrofiber-reinforced concrete with GO treatment (FRC/GO). (**d**) Elements of the spectra for FRC/GO.

**Table 1 polymers-16-01168-t001:** Characteristics provided by manufacturers of the macrofibers used.

	Macrofiber A	Macrofiber B
Polymer	Polypropylene	Polypropylene
Tensile strength (MPa)	379.1	640
Modulus of elasticity (GPa)	5.2	12
Length (mm)	48	48
MFR (g/10 min)	10.54 ± 0.24	0.52 ± 0.01
Application	Reinforcement in concrete	Reinforcement in concrete

**Table 2 polymers-16-01168-t002:** Consumption of materials for molding the test specimens.

Mixtures	Cement (g)	Natural Sand (g)	Crushing Sand (g)	Water (g)	SP (g)
FRC	1011.28	926.13	617.42	404.43	9.15
FRC/GO	1011.28	926.13	617.42	404.43	9.15

FRC = macrofiber-reinforced concrete; FRC/GO = macrofiber-reinforced concrete/GO; SP = superplasticizer.

## Data Availability

Data are contained within the article.

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
