# Peer review of "Effect of Graphene Oxide Surface Deposition Process on Synthetic Macrofibers and Its Results on the Microstructure of Fiber-Reinforced Concrete"

_polymers, 2024, doi:10.3390/polym16081168_

Round 1
Reviewer 1 Report
Comments and Suggestions for Authors
Comments
In the manuscript entitled, ‘Effect of graphene oxide surface deposition process on synthetic macrofibers and its results on the microstructure of fiber-reinforced concrete’, the author discuss about the inclusion of GO on synthetic macrofibers and its effect on FRC. I do have some queries that are given as comments below,
Sub-heading 2.6 is not understandable. Make it clear.
It will be better to indicate the structural units (number of polypropylene units) or general formula for both the macrofibers A and B.
In the SEM images (Figures 2 and 3) of the macrofibers A and B, with and without GO treatment, the treated macrofibers shows two additional structures, flake-like and thread like structure apart from the untreated fiber structure. Flake structure is due to GO. What about the thread-like structure? Moreover, this thread-like structure could be found more in treated macrofiber B. Explanation is needed.
Cross-sectional SEM images are required for the untreated and GO treated macrofibers A and B, to check the compatibility between the fibers and GO.
In a similar way, cross-sectional SEM images for the FRC and FRC/GO composites are required to get an idea about the interaction between the fibers and the cement matrix.
In figures 7a and c, the numbering from 1-10 inside the image is not clear. Replace the figures with legible numbers.
As the topic of the manuscript describes about the effect of GO, but only one composition of GO (3.2 g) was added to the fibers. The SEM images clearly shows the agglomeration of GO both in the macrofibers and in the FRC. Better to check the effect of GO with additional compositions (minimum of 2 compostion of GO), less than 3.2 g. The lower composition can avoid agglomeration issues and improve the property of the composites.
The study shows only the morphological changes, but the final property of the materials should be given more importance. The tensile and modulus of elasticity of treated macroporous fiber A, treated macroporous fiber B, FRC and FRC/GO should be analyzed to determine how the addition of GO improves the final property of the material.

Author Response
Dear editor and reviewers,
We are forwarding the manuscript with the necessary adjustments based on your comments. We would like to thank you for the indications and suggestions to improve the paper. We hope to meet the expectations based on the comments below.
Reviewer 1:
Adjustments and insertions are highlighted in this way in the new version of the paper.
- Sub heading 2.6 is not understandable. Make it clear.
The sub heading 2.6 has been rewritten as indicated.
- It will be better to indicate the structural units (number of polypropylene units) or general formula for both the macrofibers A and B.
Both fibers presented the same density value (0.91 g/cm3). However, the authors opted to add in the manuscript the mechanical properties of each fiber to better compare its mechanical performance when they are added in concrete.
- In the SEM images (Figures 2 and 3) of the macrofibers A and B, with and without GO treatment, the treated macrofibers shows two additional structures, flake like and th read like structure apart from the untreated fiber structure. Flake structure is due to GO. What about the thread like structure? Moreover, this thread like structure could be found more in treated macrofiber B. Explanation is needed.
The explanation was added and highlighted next to figures 2 and 3.
- Cross-sectional SEM images are required for the untreated and GO treated macrofibers A and B, to check the compatibility between the fibers and GO.
The aim was to evaluate the deposition of graphene oxide on the longitudinal profile of the macrofiber, considering that it will be the largest surface in contact with the concrete matrix. The presentation method followed the guidance of different authors who evaluated the fiber in the longitudinal direction. Some authors:
- https://doi.org/10.1016/j.cemconres.2019.105899
- https://doi.org/10.1016/j.compositesb.2019.107010
- https://doi.org/10.1016/j.conbuildmat.2018.08.170
- https://doi.org/10.1016/j.carbon.2020.08.051
- In a similar way, cross sectional SEM images for the FRC and FRC/GO composites are required to get an idea about the interaction between the fibers and the cement matrix.
The micrograph presented in Figure 6 (a) shows the cross-section of the untreated macrofiber and the matrix, while Figure 6 (d) shows the cross-section of the macrofiber treated with graphene oxide and the matrix.
- In figures 7a and c, the numbering from 1 10 inside the image is not clear. Replace the figures with legible numbers.
Thanks, the replacement was carried out.
- As the topic of the manuscript describes about the effect of GO, but only one composition of GO (3.2 g) was added to the fibers. The SEM images clearly shows the agglomeration of GO both in the macrofibers and in the FRC. Better to check the effect of GO with additional compositions (minimum of 2 composition of GO), less than 3.2 g. The lower composit ion can avoid agglomeration issues and improve the property of the composites.
The text was improved and highlighted next to item 2.5.
The amount of GO was used following the GO mass/fiber mass ratio recommended for carrying out the treatment on microfibers, following the literature:
LU, Lingchao; ZHAO, Piqi; LU, Zeyu. A short discussion on how to effectively use graphene oxide to reinforce cementitious composites. Construction And Building Materials, [S.L.], v. 189, p. 33-41, nov. 2018. Elsevier BV. http://dx.doi.org/10.1016/j.conbuildmat.2018.08.170;
LU, Zeyu; YAO, Jie; LEUNG, Christopher K.y.. Using graphene oxide to strengthen the bond between PE fiber and matrix to improve the strain hardening behavior of SHCC. Cement And Concrete Research, [S.L.], v. 126, p. 105899, dez. 2019. Elsevier BV. http://dx.doi.org/10.1016/j.cemconres.2019.105899.
Considering that this is the first stage of study and new treatment conditions will be listed in future studies.
- The study shows only the morphological changes, but the final property of the materials should be given more importance. The tensile and modulus of elasticity of treated macroporous fiber A, treated macroporous fiber B, FRC and FRC/GO should be analyzed to determine how the addition of GO improves the final property of the material.
Mechanical tests including the pullout resistance of macrofibers treated and not treated with graphene oxide and this effect on the shear stress of the samples were carried out and submitted to another journal. However, we can say that the treatment carried out with graphene oxide, which affected the microstructure presented in this work, contributed to significant gains in shear stress.

Reviewer 2 Report
Comments and Suggestions for Authors
In the SEM-FEG analysis, you mentioned the presence of granular elements on the surface of macrofiber A. Could you discuss the potential implications of these granular elements on the surface treatment process and the subsequent performance of the macrofiber in concrete reinforcement?
The micrographs of macrofiber B after treatment showed wrinkles on the fiber surface. How do these wrinkles affect the adhesion between the fiber and the cement paste in fiber-reinforced concrete, particularly in terms of mechanical properties and durability?
In the EDS analysis of macrofiber A, you noted differences in the surface composition between treated and untreated samples. Could you discuss how these differences in surface composition impact the chemical bonding between the macrofiber and the cement matrix, and consequently, the mechanical properties of the reinforced concrete?
The presence of small grooves on the surface of macrofiber B was highlighted. How do these small grooves influence the deposition of graphene oxide during the treatment process, and how do they contribute to the mechanical properties of the fiber-reinforced concrete?
Clarify how the observed fraying of macrofibers in the SEM-FEG micrographs of untreated concrete samples affects the mechanical behavior of the reinforced concrete, especially in terms of crack propagation and post-cracking behavior.
You mentioned that the presence of graphene oxide on the macrofiber surface led to better fiber-matrix interaction in concrete. Can you provide a detailed explanation of the mechanisms through which graphene oxide enhances the adhesion between the fiber and the cement paste?
How do the results obtained from the SEM-FEG and EDS analyses align with previous studies on fiber-reinforced concrete and the incorporation of nanomaterials? Are there any notable differences or similarities in the findings?
Considering the observed densification of elements in the concrete matrix, how do these changes correlate with the mechanical properties of the reinforced concrete? Additionally, could you discuss the potential long-term effects of these changes on the durability and service life of the concrete structures?
I suggest incorporating the following research into the introduction section to offer further perspectives on the utilization of fibers in various concrete formulations, thereby enhancing the foundational comprehension of this topic.
-Property Assessment of High-Performance Concrete Containing Three Types of Fibers
-The influence of basalt fiber on the mechanical performance of concrete-filled steel tube short columns under axial compression
-Utilization of antimony tailings in fiber-reinforced 3D printed concrete: A sustainable approach for construction materials
-Abrasion performance and failure mechanism of fiber yarns based on molecular segmental differences
Comments on the Quality of English LanguageI can see some minor grammatical errors. So carefully check the whole text.
Author Response
Dear editor and reviewers,
We are forwarding the manuscript with the necessary adjustments based on your comments. We would like to thank you for the indications and suggestions to improve the paper. We hope to meet the expectations based on the comments below.
Reviewer 2:
Adjustments and insertions are highlighted in this way in the new version of the paper.
- In the SEM-FEG analysis, you mentioned the presence of granular elements on the surface of macrofiber A. Could you discuss the potential implications of these granular elements on the surface treatment process and the subsequent performance of the macrofiber in concrete reinforcement?
Thanks, the explanation has been carried out and highlighted.
- The micrographs of macrofiber B after treatment showed wrinkles on the fiber surface. How do these wrinkles affect the adhesion between the fiber and the cement paste in fiber-reinforced concrete, particularly in terms of mechanical properties and durability?
The explanation was made after Figure 3 and highlighted.
- In the EDS analysis of macrofiber A, you noted differences in the surface composition between treated and untreated samples. Could you discuss how these differences in surface composition impact the chemical bonding between the macrofiber and the cement matrix, and consequently, the mechanical properties of the reinforced concrete?
The explanation was inserted after Figure 4 and highlighted.
- The presence of small grooves on the surface of macrofiber B was highlighted. How do these small grooves influence the deposition of graphene oxide during the treatment process, and how do they contribute to the mechanical properties of the fiber-reinforced concrete?
The explanation was inserted after Figure 5 and highlighted.
- Clarify how the observed fraying of macrofibers in the SEM-FEG micrographs of untreated concrete samples affects the mechanical behavior of the reinforced concrete, especially in terms of crack propagation and post-cracking behavior.
Thanks, the explanation was inserted after Figure 6 and highlighted.
- You mentioned that the presence of graphene oxide on the macrofiber surface led to better fiber-matrix interaction in concrete. Can you provide a detailed explanation of the mechanisms through which graphene oxide enhances the adhesion between the fiber and the cement paste?
The explanation was inserted after Figure 7 and highlighted.
- How do the results obtained from the SEM-FEG and EDS analyses align with previous studies on fiber-reinforced concrete and the incorporation of nanomaterials? Are there any notable differences or similarities in the findings?
A discussion was done and highlighted below Figure 6, next to paragraphs 5 and 6.
- Considering the observed densification of elements in the concrete matrix, how do these changes correlate with the mechanical properties of the reinforced concrete? Additionally, could you discuss the potential long-term effects of these changes on the durability and service life of the concrete structures?
Inserted and highlighted the comment below Figure 7.
- I suggest incorporating the following research into the introduction section to offer further perspectives on the utilization of fibers in various concrete formulations, thereby enhancing the foundational comprehension of this topic.
-Property Assessment of High-Performance Concrete Containing Three Types of Fibers
-The influence of basalt fiber on the mechanical performance of concrete-filled steel tube short columns under axial compression
-Utilization of antimony tailings in fiber-reinforced 3D printed concrete: A sustainable approach for construction materials
-Abrasion performance and failure mechanism of fiber yarns based on molecular segmental differences
Thank you for the suggestions, they were used and highlighted in the introduction of the work.

Round 2
Reviewer 1 Report
Comments and Suggestions for Authors
Comments
How do the macrofiber A and macrofiber B differ in structure? Mention their structure or add a sentence about it.
Author Response
We added the melt flow rate values for both fiber in Table 1. Based on these results macrofiber B presented higher viscosity than macrofiber A, possibly due to its higher molar mass. It is also possibly that the differences observed in the mechanical properties for both fiber can be related to this difference in molar mass, since both fibers are composed only of polypropylene. The changes are highlighted in blue.
Reviewer 2 Report
Comments and Suggestions for Authors
After thoroughly reviewing the revised manuscript, I propose accepting it at this stage.
Author Response
Thanks.